

# Past changes in natural and managed snow reliability of French Alps ski resorts from 1961 to 2018

Lucas Berard-Chenu[1,2], Hugues François[1], Emmanuelle George[1], Samuel Morin[2]

[1] Univ. Grenoble Alpes, INRAE, LESSEM, 38000 Grenoble, France
[2] Univ. Grenoble Alpes, Université de Toulouse, Météo-France, CNRS, CNRM, Centre d'Etudes de la Neige, 38000 Grenoble, France

*Correspondence to*: Lucas Berard-Chenu (lucas.berard@inrae.fr)

**Abstract.** Snow reliability is a key climatic impact driver for the ski tourism industry, for which there is more literature regarding future projections than past observed impacts. This study provides an assessment of past changes in natural and managed snow cover reliability from 1961 to 2018 in the French Alps. In particular, we used snowmaking investment figures to infer the evolution of snowmaking coverage at the ski resort scale for 16 ski resorts in the French Alps. We find different benefits of snow management to reduce the variability and long term decrease in snow cover reliability because of the heterogeneity of the snowmaking deployment trajectories across ski resorts. The frequency of challenging conditions for ski resort operation over the 1991-2018 period increased in November and February to April compared to the reference period 1961-1990. In general, snowmaking had a positive impact on snow reliability, especially in December to January. While for the highest elevation ski resorts, snowmaking improved snow reliability for the core of the winter season, it did not counterbalance the decreasing trend in snow cover reliability for lower elevation ski resorts and in the spring.

## 1. Introduction

Ski tourism is a major socio-economic component of mountain areas mostly in developed countries (Vanat, 2020). Due to its reliance on the snow cover, ski tourism has for long been identified to be particularly vulnerable to climate change impacts, such as decreases in snow cover duration at low elevation in mountain regions (Martin et al., 1994, Beniston et al., 1995, Abegg et al., 2006). In order to reduce the impact of natural snow cover variability and its long term decreasing trend due to climate change, snowmaking has emerged and is nowadays routinely used in almost all ski resorts in developed countries (Steiger et al., 2019). Recent progress has been made in the literature to better account for snowmaking in future projections of snow reliability in ski resorts in Europe at the pan-European scale (Morin et al., 2021), in separate European countries (Austria: e.g. Marke et al., 2015; France: e.g. Spandre et al., 2019; Norway: e.g. Scott et al., 2020a) and North America (Scott et al. 2020b). However, while several studies have documented strong reductions in snow cover amount, depth and duration in many mountain regions of the world over the past decades (Mote et al., 2018, Klein et al., 2016, Marty et al., 2017, Matiu et al., 2021, Hock et al., in press), explicit assessments of the impact of climate change on ski resorts operations,



based on past observations, have remained limited (Beaudin & Huang, 2014, Hamilton et al., 2003). In fact, the emergence in past decades and popularization in major news media of several studies addressing future climate change risks to ski tourism, together with the recurrence of snow-scarce winters in Europe (Durand et al., 2009) and North America (Cooper et al., 2016), have somehow lead to the broadly accepted consideration that climate change has indeed already been having a
strong impact on ski resorts operations in the past and at present (Knowles and Scott, 2020). Based on interviews with ski resort managers and ski tourists, several studies have documented the fact that many stakeholders from the ski tourism industry have perceived a change in meteorological and snow conditions, which they did not always attribute to climate change (Trawöger, 2014). The strongest evidence on climate change impacts to ski tourism has therefore been inferred primarily from future climate change projections, which is then used to interpret past and present situations, in the absence of
solid studies assessing impacts based on past observations. In a way, the link between climate change and snow cover reliability appears to be so direct and obvious (less snow in a warmer world) that it has been somehow neglected in previous studies addressing past changes, with only a few exceptions. Alongside this knowledge gap, the efficiency of snowmaking based on past observations has seldom been assessed quantitatively: this also requires in-depth analysis of ski resorts operating conditions for past seasons, in a way that allows disentangling the contribution of snowmaking to actual snow
conditions.

Previous studies have led to the development of a sophisticated modelling system to simulate snow conditions in French ski resorts for the past and future, taking explicitly into account grooming and snowmaking, based on the meteorological reanalysis system SAFRAN, the snow cover model Crocus and accounting for several key features of individual ski resorts (Spandre, 2016b, Spandre et al., 2019). This model chain takes as input, for each ski resort, the fraction of ski slopes covered
with snowguns. However, in past studies the time evolution of snowmaking was applied uniformly to all ski resorts (Spandre et al, 2019), in the absence of resort-level data describing the past time evolution of snowmaking fractional coverage in ski resorts. A proper assessment of the impact of the interannual variability and long term climate change on ski resorts operating conditions requires not only to take into account observed meteorological conditions driving the time variations of snow conditions in ski resorts, but also the time evolution of their individual snowmaking capacity. In the French Alps,
snowmaking has emerged in the 1990s, but not all ski resorts have followed the same development pathway in terms of snowmaking, and there is no consolidated database describing this past evolution (Berard-Chenu et al., 2020). For a given change in a climatic impact driver operating at the scale of an entire mountain range, various snowmaking equipment pathways in different ski resorts are thus anticipated to lead to different impacts.

In this study, we quantitatively assessed the changes in operating conditions of ski resorts, with and without snowmaking,
taking into account the variability in snowmaking development dynamics across ski resorts. To do so, we analysed snowmaking investments figures by 16 ski resorts in the French Alps from 1997 to 2018 and developed an original method to infer the time evolution of the snowmaking coverage for each ski resort. This data was used to feed the SAFRAN-Crocus



model chain, making it possible to perform simulations with and without snowmaking for the past decades. Indeed, there is no long-term record of snow conditions in ski resorts, which could be used for this purpose, and there is no way to reconstruct what would be the snow cover situation without snowmaking, once it has been implemented in actual ski resorts. Numerical simulations make it possible to compare the state of the snow cover with and without snowmaking, everything else being equal. Section 2 introduces the methodology used for this study, section 3 provides the results, section 4 discusses them while section 5 concludes.

## 2. Material and methods

In the absence of a consolidated database providing the time evolution of the snowmaking coverage in each ski resort, this study uses investment figures related to snowmaking investment, in order to infer the time evolution of snowmaking coverage in individual ski resorts in the French Alps. To the best of our knowledge, this approach is unprecedented in the scientific community. Such estimates are then used to process snow cover simulations produced using the SAFRAN-Crocus model chain, with and without snowmaking, spanning the time period from 1961 to 2018, enabling us to quantify past changes in snow reliability at the ski resorts scale, explicitly accounting for snowmaking equipment dynamics in individual ski resorts.

### 2.1. Snowmaking investments dataset

Investment figures come from a snowmaking investment data set for 100 French Alps ski resorts from 1997 to 2018 (Berard-Chenu et al., 2020). The data set originates from the professional journal 'Montagne Leaders' who manages a yearly investment survey filled in a declarative manner by ski lift operators. Although collected by a non-scientific and unofficial organization, these data hold significant value. For instance Falk and Vanat (2016) and Berard-Chenu et al. (2020) used this data source for academic research purposes. The investment survey covers 5 types of investment: new ski lift, ski lift maintenance, snowmaking, ticketing, and ski slope remodelling. Each type of investment is broken down into subgroups. Snowmaking investments are decomposed into 6 types: equipment and concepts, water reservoirs, electricity, civil engineering, buildings and other. In our investment dataset we do not consider all snowmaking investments but only those referring to "equipment and concepts". Indeed, the subgroup of "equipment and concepts" is the most relevant proxy to capture the growth rate of the snowmaking coverage of ski resorts as it only accounts for the purchase of snow guns. The other snowmaking investments can refer to an increase of the snowmaking instantaneous production capacity i.e. an electrical power increase or an expansion of the water storage capacity. These investments may increase the snowmaking production efficiency without any growth of the ski slope areas equipped with snowmaking facilities. The investment dataset only contains ski resorts which have invested at least once in snowmaking "equipment and concepts" over the 1997-2018 period. Investments are in constant prices: we derived constant price investments from current price investments with the GDP deflator, a widely used inflation adjustment index.





The literature points to a broad range of figures on cost investment for snowmaking facilities. Table 1 shows several
estimates related to snowmaking investment costs per unit ski slope surface area, ranging from 28k€ to 240k€ per hectare. In
France, the latest estimate from *Domaines Skiables de France* dates back to 2018, in an internal training document intended
for snowmakers, with a unit cost ranging between 130 to 150k€ per hectare.

**Table 1: Snowmaking investment cost to equip ski slopes area**

| Source | Value mentioned | Value in constant price from 2018 (k€ per hectare) |
|---|---|---|
| SEATM (1989) | 1 million FRF per hectare | 240 |
| Vlés (1997) | 1 million FRF per hectare | 200 |
| CIPRA (2004) | 136 k€ per hectare | 162 |
| Abegg et al. (2007) | 25 to 100 k€ per hectare | 28 to 114 |
| Breiling et al. (2008) | 150 k€ per hectare | 166 |
| Badre et al. (2009) | 150 to 200k€ per hectare | 165 to 220 |
| DSF (2018) | 130 to 150k€ per hectare | 130 to 150 |


### 2.2. Snowmaking fractional coverage reference dataset

Atout France, which is the national organization responsible for monitoring the tourism sector, has not published estimations
of snowmaking facilities since 2009. These estimates were the only reference for professionals (Badré et al., 2009), until the
study of Spandre et al. (2015). The latter assessed that 32% of ski slope areas in the French Alps were equipped with
snowmaking facilities in 2014, based on a survey involving ski resorts managers. They also predicted that this proportion
was likely to reach 43% by 2020. DSF [*Domaines Skiables de France*], the professional association of the French ski resorts
operators, produces a snowmaking coverage average rate based on information collected from its members on an annual
basis. In its most recent release in 2020, it indicated that 37% of ski slope areas in France were equipped with snowmaking
facilities. The comparison between the DSF snowmaking facilities rate in 2020 and the estimation of Spandre et al. is
difficult since the latter only considers the French Alps and it is not fully certain that the mean indicator provided by DSF is
representative of the French ski industry.

In contrast to the national situation lacking quantitative estimates of snowmaking coverage for individual ski resorts, such
information is available for a subregion of the northern French Alps (*Département de la Savoie*) which corresponds to NUTS



3-level in the European union nomenclature of territorial units. Indeed, the *Direction Départementale des Territoires de la*
*Savoie*, an administration in charge of the water policy, has initiated an observatory of snowmaking. This public body
collects each year snowmaking data from ski resorts in Savoie, among which the snowmaking fractional coverage. Among
the 32 ski resorts equipped with snowmaking facilities and registered in the *Direction Départementale des Territoires de la
Savoie*'s database, we removed 3 ski resorts with missing or questionable data. We have a sample of 29 ski resorts after data
cleaning. For each ski resort, information is available for their total ski slopes area and its area equipped with snow
production facilities for the years 2010, 2011, 2015, 2016, 2017 and 2018.

## 2.3. Relationship between snowmaking investments and snowmaking fractional coverage

We developed a model analyzing the relationship between the snowmaking surface area coverage with snowmaking
investments. Among various possibilities, we retained a model relating them in a linear way (an exponential model was also
tested, leading to insignificant differences for ski resorts with sufficiently numerous observation data).
The linear model to be estimated is:

$$SMA_t = SMA_{t-1} + \alpha(I_t - I_{t-1}) + \varepsilon$$

where t denotes the year.

The left-hand variable SMA denotes the ski slope area equipped with snowmaking facilities and I the cumulated
snowmaking investment. α represents the snowmaking unit surface area equipment cost and ε is an error-term. We estimated
the area equipped with snowmaking facilities based on the area equipped and the snowmaking investment increased both
from the previous year. Among the 29 ski resorts equipped with snowmaking equipment, 13 showed no increase in the part
of their snowmaking coverage despite the fact they made snowmaking investments over the study period. We removed these
13 ski resorts from our sampling since an increase in the snowmaking coverage rate over the six reference years is required
to estimate the α coefficient in our modelling. Thus our final sample for the evolution of the individual snowmaking
coverage encompasses 16 ski resorts.

## 2.4. SAFRAN-Crocus model chain, ski resorts geospatial modelling and snow reliability indicator

We used the SAFRAN-Crocus model chain to simulate snow cover characteristics from 1961 to 2018. SAFRAN is a
meteorological reanalysis system, combining in-situ observations and numerical weather prediction model output to provide
an estimate of the meteorological conditions as a function of elevation (by 300 m elevation steps), for "massifs", i.e.
mountain areas assumed to be meteorologically homogenous. This system makes it possible to reconstruct the time evolution
of the meteorological conditions in the mountain regions of France since the late 1950s (Durand et al., 2009, Vernay et al.,



2019), and has been used as a basis for several previous studies on the snow cover reliability of ski resorts in the French Alps and Pyrenees (e.g. Spandre et al., 2019, and references therein). The hourly resolution SAFRAN meteorological conditions are used as input to the detailed, multilayer snow cover model Crocus, resolving natural processes occurring in the snow

cover and at its interfaces with the underlying ground and upperlying atmosphere (Vionnet et al., 2012). In its Resort version (Spandre et al., 2016b), the Crocus model is also equipped with dedicated options for representing snow grooming and snowmaking, both in terms of their physical characteristics but also typical management operations timing and snowmaking production rate (Spandre et al., 2016a, 2019 Hanzer et al. 2020, Morin et al., 2021). Crocus model runs were carried out for each relevant SAFRAN massif and all elevations, not only on flat terrain but also for 8 main orientations (N, NE, E, SE, S,

SW, W, NW) and slope angles of 10, 20, 30 and 40°. Simulation outputs were then aggregated for individual ski resorts at a daily scale used to calculate a reliability index for different periods along the ski season. For each ski resort, a ski resort gravitational envelope is computed based on a Digital Elevation Model (DEM) and the location of ski lifts (François et al., 2014, 2016, Spandre et al., 2019). Depending on the snowmaking coverage, simulations carried out with and without snowmaking are combined, as described in Spandre et al. (2019). The location of ski slopes equipped with snowmaking is

determined based on their position within the ski resort (elevation, distance to major ski lift, distance to main housing infrastructure), following Spandre et al. (2016a). This entire procedure enables to generate, for a given snowmaking coverage, a resort-level snow cover indicator, quantifying the fraction of the ski resort surface area where the amount of snow exceeds a given threshold (here, 100 kg m$^{-2}$ snow water equivalent, which corresponds to 25 cm of snow with a density of 400 kg m$^{-3}$). Daily values of this indicator are averaged for each month from November to April, as well as for the

Christmas time period from December 20 to January 5. This approach is comparable to the "snow reliable skiing terrain" approach developed and used by Steiger and Stötter (2013) and Scott et al. (2019). We also computed the combined reliability index, corresponding to the weighted average of the Christmas reliability indicator (15%) and the February reliability indicator (85%), shown by Spandre et al. (2019) to correlate strongly, at the scale of the entire French Alps, to annual skier visits from 2002 to 2014, when using a fix 30% snowmaking coverage for all ski resorts. This method provides,

ultimately, resort-level snow reliability indicators at the monthly scale (plus Christmas time period and combined indicator) spanning the full time period from 1961-1962 season to 2018-2019 season continuously, whose values can be computed depending on the snowmaking fractional coverage value, with a regular 15% step between 0% and 90% where 0% corresponds to grooming only. The time variable snowmaking coverage where considered as a linear interpolation of the values obtained by steps of 15% fractional coverage. In addition, based on the gravitational envelope computed for each ski

resort, we compute the fraction of the ski resort above 2000 m elevation. This elevation threshold roughly corresponds to the divide between low-elevation and high-elevation mountain areas, most relevant in the context of elevation-dependent natural snow cover trends (Matiu et al., 2021) in the European Alps. It enables us to compare ski resorts' geographical characteristics, not only in terms of their minimum/maximum or mean elevation, but using a single indicator representative of their elevational coverage.





## 2.5. Statistical analysis of snow reliability indicators

The annual values of the annual-scale indicators values were post-processed as follows. Indeed, the study focuses on ski resorts operations and how they have been modified in the past decades, accounting for the role of snowmaking. What matters most for ski resorts is not the mean multi-annual snow reliability, or other metrics characterizing their multi-annual average conditions, but rather how frequently challenging operating conditions are encountered. Indeed, snow scarce

winters, primarily related to the interannual variability of meteorological and snow conditions, and influenced by long term climate trends, are a key concern to the ski tourism industry (Abegg et al., 2020, Morin et al., 2021). We therefore focus on the characteristics of challenging winter seasons, and whether their frequency has changed in time. We take as a reference the value of the 20th quantiles of snow reliability indicators for the 1961-1990 reference time period (referred to as Q20 values), which correspond to the one-in-five years typical challenging snow conditions and is relevant for ski resorts

operators. This value depends on the ski resort (because of their various elevation and elevation span), on the snowmaking fractional coverage, and on the time period of the year. We then analyze, for the time period 1991-2018, the frequency of winter seasons for which the snow reliability values are below the reference Q20. If the value corresponds to more than 20% of the years (i.e., 6 years in the case of a full 30 years time period), this means that the snow conditions have worsened compared to the reference period, and vice versa. This approach enables fair comparisons across time, for a given ski resort,

without comparing, in absolute terms, the mean and low-quantile values characterizing snow conditions in different ski resorts or with different snow management options. The absolute values for mean or Q20 snow reliability values for different ski resorts can be very different, and need to be interpreted separately in assessments at the scale of ski resorts, together with the business model of the tourism destination encompassing each ski resorts, within which the reliability of the snow cover on ski pistes can play a very different role. For example, large high elevation ski resorts are based on high reliability values

with low interannual variability, and offer little alternatives to downhill skiing in case of ski resorts closure, while lower elevation ski resorts embedded in a diversified destination are less dependent on a reliable snow cover on ski pistes, which is often lesser guaranteed in such a case (Luthe et al., 2012; George-Marcelpoil & François, 2016). Figure 1 presents a schematic description of the method and the data processing.





**Figure 1.** Schematic description of the method

## 3. Results

### 3.1. Relationship between snowmaking investments and fractional coverage

The linear modelling for reconstructing snowmaking coverage, described in section 2.3, was applied to 16 ski resorts in Savoie over the observed period (2010-2018). Figure 2 shows the estimated evolution of the snowmaking coverage rate over 1997-2018 for these 16 ski resorts. Estimates of snowmaking coverage are derived from individual investments of each ski resort. The figure highlights the heterogeneity of the snowmaking deployment trajectories depending on ski resorts. Investment choices directly affect the increase of the snowmaking coverage for each ski resort: the snowmaking coverage



rates range from 0 to 37% in 1997 and from 14% to almost 70% in 2018. The average of the evolution of the snowmaking coverage based on the 16 ski resorts (black line) is higher than the national average (dashed line) throughout the period considered. The figure shows that a larger difference is reached for the year 2006 with a difference of almost 10% while in 2018 or 1997 the difference is 2% and 5%, respectively.

The linear modelling estimated a mean alpha coefficient of 8.48 $10^{-3}$, which corresponds to 118k€ per hectare with $CI_{95\%} =$ [87, 180]. A bootstrap method (n = 1000 re-sampling of coefficient values) was applied to provide the 95% confidence interval for the coefficient estimate.

**Figure 2.** Evolution of the snowmaking coverage rate over 1997-2018 for the 16 ski resorts in Savoie with the linear model. Colours represent individual ski resorts, anonymized and ordered as a function of decreasing fraction of the domain below 2000 m.

### 3.2. Time evolution of combined (Christmas and February) snow reliability and relationship to snowmaking coverage



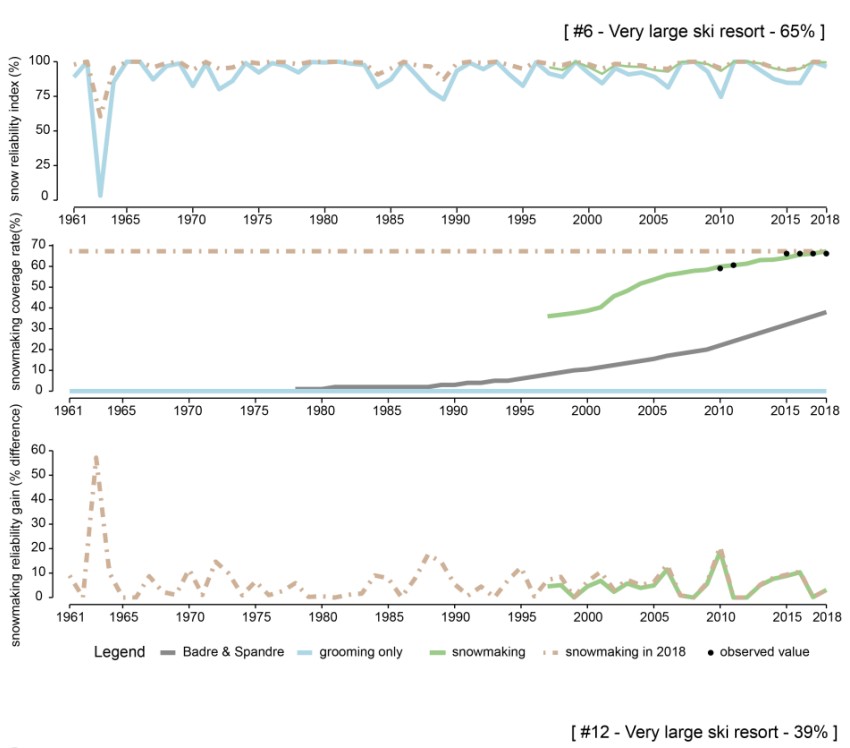

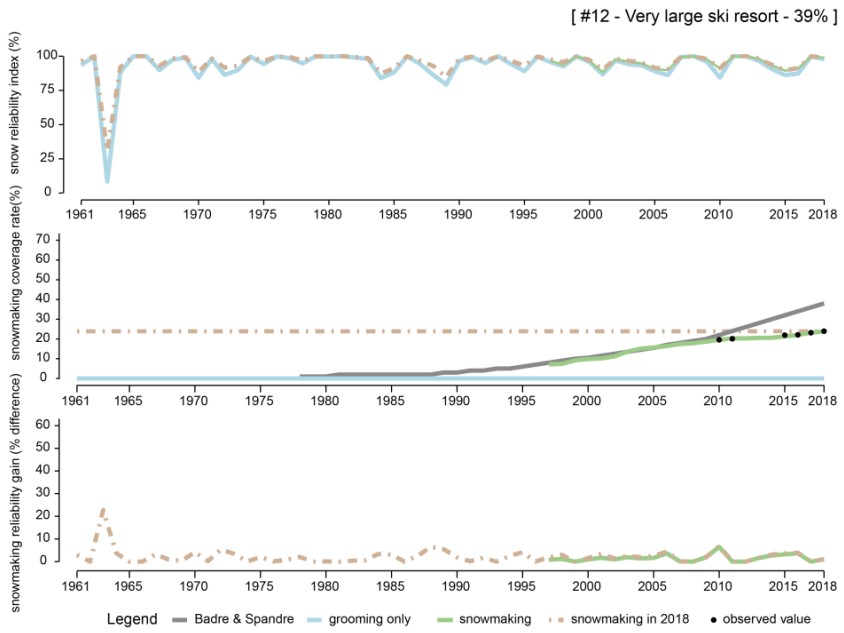

**Figure 3.** Time evolution of the combined snow reliability indicator and snowmaking fractional coverage for two contrasting very large ski resorts (ski resort #6 has 65% of its ski area below 2000m, 39% for ski resort #12) in Savoie from 1961 to 2018. The top row represents the time variations of the indicator for several snow management options (grooming only, snowmaking coverage corresponding to Badré et al.,2009 - mean value for al French ski resorts, individual evolution of the snowmaking coverage based on investment figures, and situation corresponding to using the snowmaking coverage in 2018 for the whole time period). The bottom row indicates the corresponding snowmaking fraction coverage. The last row represents the increase in snow cover reliability compared to the grooming only situation. The ski resort #6 shows a higher-than-average snowmaking equipment, while the ski resort #12 shows a lower-than-average snowmaking equipment.





Figure 3 displays the time evolution of the combined snow reliability indicator and snowmaking fractional coverage for two contrasting very large ski resorts in the Savoie from 1961 to 2018, with a snowmaking coverage below (reaching ca 24% en 2018) and above (ca 67% in 2018) the national average (estimated of the order of 38% in 2018), respectively. While some

inter-annual variability is observed and both ski resorts display contrasting geographical characteristics regarding the share of their ski area below 2000m, in general the snow reliability values have remained rather high throughout the entire time period, except for the peculiar winter 1963-1964. Compared to a situation without snowmaking, the various snowmaking dynamics lead to very similar snow reliability values, with either no increase in reliability (when the reliability with groomed snow only is already very high and close to 100%) or an increase on the order of 5 to 10%. In this example, the ski resort #6

with the larger snowmaking coverage fraction shows lower interannual variability, when snowmaking is taken into account, than the ski resort #12 with lower snowmaking fractional coverage.

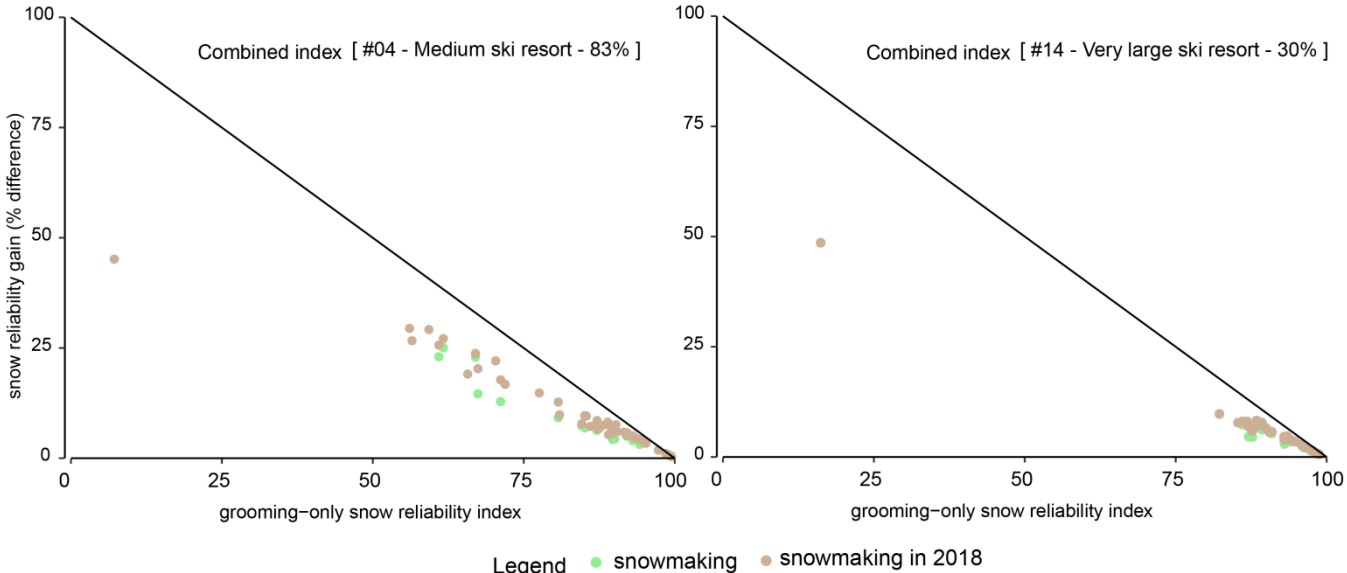

**Figure 4.** Relationship between the combined snow cover reliability difference (y-axis) between configurations with snowmaking and the
snow cover reliability with grooming only (no snowmaking, x-axis), for two contrasting ski resorts.

Figure 4 shows the relationship between the annual snow cover reliability difference between configurations with snowmaking and the snow cover reliability without grooming (no snowmaking, x-axis), for two contrasting ski resorts (lower than average snowmaking coverage with 83% of the ski area below 2000 m on the left and higher than average snowmaking coverage on the right and with 30% of the ski area below 2000 m). It shows that, in these cases, the natural

snow cover deficit can almost be fully compensated, for this indicator focusing on the Christmas and February time periods with high snowmaking coverage, except in the cases where the deficit is too large (in particular, the winter 1963-1964).

### 3.3. Time evolution of monthly snow reliability indicators



**Figure 5.** Time evolution of the combined snow reliability indicator and snowmaking fractional coverage for ski resort #04 (medium size, 83% of its surface area below 2000 m elevation) from 1961 to 2018. The left column represents the time variations of the indicator for several snow management options (grooming only, individual evolution of the snowmaking coverage based on investment figures, and the situation corresponding to using the snowmaking coverage in 2018 for the whole time period). The right column represents the increase in snow cover reliability compared to the grooming only situation (taking into account the evolution in snowmaking coverage since 1997 or using the fractional coverage for 2018 throughout the entire record). Each row corresponds to a monthly snow reliability indicator: December to March.





**Figure 6.** Similar to Fig. 5 but for ski resort #14 (very large ski resorts, 30% of the surface area below 2000 m elevation)

Figure 5 and 6 show the time evolution of the monthly snow reliability indicators for the two contrasting ski resorts introduced above, showing the large differences between reliability indicator values across the different months, in particular the lower values at the beginning and end of the winter season, and peak values, on average, in January and February.



**Figure 7.** Relationship between the monthly (December, January, February and March), from top to bottom) snow cover reliability difference (y-axis) between configurations with snowmaking and the snow cover reliability with grooming only (no snowmaking, x-axis), for two contrasting very large ski resorts and for the time period from 1961 to 2018, on the left with below average snowmaking coverage, and on the right with above-average snowmaking coverage.

Figure 7 shows the relationships between the monthly indicator values in December to March, for the same contrasting ski resorts in terms of their snowmaking coverage. It clearly outlines the differences in snow reliability between months and between ski resorts, and how snowmaking can (or cannot) compensate for the natural snow cover deficit, depending on the



meteorological conditions but also on the snowmaking fractional coverage of individual ski resorts, which depend on individual strategic choices (and the means to implement them). The gap between snow reliability gains for the two contrasting ski resorts are larger, regardless of the snowmaking configuration, in the beginning (December and January) rather than the peak (February and March) of the winter season. For the December index, there is a maximum gain below

25% for the ski resort #4 (with lower than average snowmaking coverage and 83% of its ski area below 2000 m) while the ski resort #14 (higher than average snowmaking coverage and 30% of its ski area below 2000 m) can benefit from up to 50% snow reliability gains. With a snow coverage rate corresponding to 54% in 2018 (brown dots), ski resort #14 would have increased its snow reliability with snowmaking almost always up to 25% when its grooming-only snow reliability went down below 50%.

**3.4.   Change in the frequency of challenging snow conditions for the 16 ski resorts in Savoie**

In this section we analyze the change in the frequency of occurrence of challenging snow conditions from 1961 to 2018, by splitting the entire time period in two periods of approximately 30 years (1961-1990 and 1991-2018) and analyzing how frequently, in the second time period, the snow conditions were better or worse than the Q20 threshold of the reference period 1961-1990. We provide the results for the 16 ski resorts in Savoie for which a dedicated analysis of the snowmaking

coverage evolution was carried out.





**Figure 8.** Frequency of occurrence of monthly snow reliability challenging conditions, between 1991 and 2018, below the reference Q20 value for the time period 1961-1990, without snowmaking throughout the entire time period. Each row corresponds to a ski resort (ranked as a function of decreasing fraction of the domain below 2000 m). Each column corresponds to a monthly reliability indicator. The values in the matrix correspond to the Q20 reference snow reliability indicator. Red cells correspond to larger frequencies during 1991-2018 than during the reference time period (more often challenging conditions), and vice versa in blue.

Figure 8 shows a synthesis, for the 16 ski resorts of our sample, of the monthly reliability values (reference Q20, which characterizes the upper threshold of challenging snow conditions) and the evolution of the frequency of exceedance below this threshold, in a situation without snowmaking. Largest snow reliability values for the reference period are found in February and March, with Q20 values ranging from 86 to 100% in February and from 90 to 100% in March. Snow reliability values in November are low, on the order of 0 (grey cells) to 10% maximum, increasing through December and January to February and March and lower values in April. The Figure shows that the months most affected by a change from 1961-1990 to 1991-2018 are in November (starting from low values), and February to April, with increasing rates of increases in the frequency of challenging conditions progressing with the month (February through April) as a function of ski resorts





elevations (largest and highest elevation ski resorts show changes mostly in November and April, with changes in a growing number of months as the elevation range starts from lower elevation). The rate of change for the Christmas time period and January shows rather a decrease in the frequency of snow conditions below the reference value. This result is fully consistent with the analysis of natural snow cover trends in the European Alps (Matiu et al., 2021), showing contrasting patterns depending on the elevation (represented here through the fraction of the ski resort below 2000 m) and on the month of the

year and marked decreases in snow indicators especially at low to intermediate elevations throughout the winter season, and for ski resorts positioned at higher elevation mostly during springtime.

| | November | December | Christmas | January | February | March | April |
|---|---|---|---|---|---|---|---|
| #01 ( M ) 100 % | 0 | 22.7 | 33.4 | 62.6 | 99.9 | 99.8 | 87.3 |
| #02 ( L ) 99 % | 0.2 | 21.1 | 37.2 | 73.9 | 99.7 | 99.7 | 93 |
| #03 ( L ) 97 % | 0 | 27.9 | 37.9 | 75.3 | 99.2 | 99.7 | 94.3 |
| #04 ( M ) 83 % | 0.6 | 26.4 | 44.5 | 75.3 | 99.7 | 99.7 | 92.7 |
| #05 ( M ) 73 % | 3.3 | 52.8 | 68.8 | 81.7 | 93.9 | 98.1 | 84.4 |
| #06 ( XL ) 65 % | 3.8 | 57.8 | 75 | 88.8 | 99.3 | 99.8 | 94.7 |
| #07 ( L ) 64 % | 4.5 | 56.2 | 74.8 | 92.1 | 100 | 100 | 98.1 |
| #08 ( XL ) 48 % | 6.7 | 56.3 | 75.7 | 90.8 | 96.6 | 97.2 | 92.8 |
| #09 ( L ) 45 % | 7.2 | 63.4 | 75.4 | 91.7 | 100 | 100 | 98.3 |
| #10 ( XL ) 40 % | 8.6 | 61.9 | 81.7 | 95.1 | 99.9 | 100 | 99 |
| #11 ( S ) 39 % | 11.1 | 67.5 | 80.9 | 95 | 100 | 100 | 98.3 |
| #12 ( XL ) 39 % | 4.2 | 43.7 | 59.4 | 85.2 | 97.5 | 98.9 | 93.2 |
| #13 ( M ) 39 % | 5.3 | 51.1 | 67 | 84.8 | 94.6 | 97.1 | 86.1 |
| #14 ( XL ) 30 % | 11.2 | 67.7 | 81.7 | 94.1 | 99.7 | 100 | 98.5 |
| #15 ( XL ) 14 % | 20.5 | 78.1 | 92 | 99.3 | 100 | 100 | 99.8 |
| #16 ( XL ) 2 % | 28.4 | 79.9 | 93 | 98.5 | 100 | 100 | 100 |

Size
No. ski resort ———— #16 ( XL ) 98 %
Fraction of the ski slope area below 2000 m.

November ———— Monthly index
9.6
Color: number of years (1991–2018) below the Q20 for the 1961–1990 period
Number: Q20 value for the 1961–1990 period

Number of years (1991–2018) below the Q20 for the 1961–1990 period
0  2  4  6  8  10  12  14  16

**Figure 9.** Same as Fig.8 but for the snowmaking coverage of 2018 throughout the entire period from 1961 to 2018 (including for the
reference time period 1961-1990).

Figure 9 shows the same results as on Fig. 8, but accounting for snowmaking with a fixed snowmaking coverage for each ski resort, corresponding to the value reached in 2018. Compared to the snow reliability values of Fig. 8 with grooming only, the





reference Q20 value, with snowmaking, is generally higher. However, with the difference of the November situation, the

general trend in the frequency of exceedance below the Q20 reference time period shows a very similar pattern than in the

natural snow situations, with a shift in snow reliability values but the same temporal pattern. This shows that the trend

towards increased snow scarcity is rather similar even with snowmaking, but starting from a higher snow reliability level.

Changes are most pronounced for the ski resorts with a larger fraction below 2000 m, and for higher elevation ski resorts

only in the beginning and end of the winter season (November and March-April), with lesser negative changes (and even

increases) in the core of the winter.

| | November | December | Christmas | January | February | March | April |
|---|---|---|---|---|---|---|---|
| #01 ( M ) 100 % | 0 | 0.1 | 1.3 | 32.4 | 99.3 | 98.8 | 68.4 |
| #02 ( L ) 99 % | 0 | 3 | 13.6 | 62.7 | 99.3 | 99.1 | 74.9 |
| #03 ( L ) 97 % | 0 | 3.9 | 10.3 | 63.8 | 98.9 | 99.2 | 90 |
| #04 ( M ) 83 % | 0 | 18.2 | 28.6 | 67 | 98.3 | 98.5 | 83.6 |
| #05 ( M ) 73 % | 0 | 3.8 | 11.9 | 46.8 | 86.6 | 97.2 | 62.3 |
| #06 ( XL ) 65 % | 0.3 | 11.7 | 25.1 | 56.1 | 96.2 | 98.4 | 81.1 |
| #07 ( L ) 64 % | 0.4 | 20.7 | 35.2 | 71.3 | 99.9 | 100 | 92 |
| #08 ( XL ) 48 % | 1.7 | 35.7 | 51.8 | 84.7 | 96.1 | 96.9 | 89.8 |
| #09 ( L ) 45 % | 0.5 | 26.2 | 45.6 | 77.1 | 100 | 100 | 95.5 |
| #10 ( XL ) 40 % | 1.3 | 25.9 | 43 | 74.2 | 99.5 | 99.9 | 94.8 |
| #11 ( S ) 39 % | 0.7 | 34.4 | 46.4 | 87.7 | 100 | 100 | 94.3 |
| #12 ( XL ) 39 % | 1.9 | 23.1 | 39.5 | 73.5 | 96.7 | 98.8 | 87.9 |
| #13 ( M ) 39 % | 0.4 | 16.3 | 29.6 | 60.2 | 86.4 | 90.6 | 69.1 |
| #14 ( XL ) 30 % | 1 | 27.3 | 42.9 | 78.3 | 99.3 | 100 | 94.7 |
| #15 ( XL ) 14 % | 5.3 | 57.5 | 80.5 | 97.7 | 100 | 100 | 99.6 |
| #16 ( XL ) 2 % | 9.6 | 56.7 | 86.3 | 97.2 | 100 | 100 | 100 |

Size ⎯⎯⎯
No. ski resort ⎯⎯⎯ #16 ( XL ) 98 %
Fraction of the ski slope area below 2000 m. ⎯⎯⎯

November ⎯⎯⎯ Monthly index
9.6
Color: number of years (1991–2018) below the Q20 for the 1961–1990 period
Number: Q20 value for the 1961–1990 period

Number of years (1991–2018) below the Q20 for the 1961–1990 period

0  2  4  6  8  10  12  14  16


**Figure 10.** Same as Fig. 8 and 9 but with a snowmaking coverage variable in time for each ski resort

Figure 10 shows the results in the same form as Fig. 8 and 9, but takes into account the fact that each ski resort had different

snowmaking coverage dynamics over the past decades. However, due to the fact that snowmaking generalized in the 1990s,

the Q20 reference values for the period 1961-1990 are roughly similar to the figures in Fig. 8. In general, the figure shows

that snowmaking had a positive impact on snow reliability in the ski resorts of this sample, especially in December to



January and for the entire winter season for the highest elevation ski resorts. For smaller and lower elevation ski resorts, snow conditions with snowmaking during the period 1991-2018 are often worse than without snowmaking between 1961 and 1990, which means that in such cases snowmaking did not counterbalance the decreasing trend in snow cover reliability.

## 4. Discussion

In this study, we have quantified the change in snow reliability in several ski resorts of the French Alps from 1961 to 2018, during a time when natural snow cover evolution has shown very strong changes especially at low to intermediate elevations, mostly attributed to the effect of atmospheric warming due to climate change (Matiu et al., 2021). Our study based on a long time period makes it possible to analyze the changes in terms of the frequency of occurrence of challenging snow conditions, defined during the reference period 1961-1990 as the value separating the 20% worst seasons (lower snow cover indicator

values) from the full record. Based on snowmaking investment figures for 16 ski resorts in Savoie (Northern French Alps) we reconstructed their snowmaking coverage trajectory since 1997, enabling us to quantify the impact of snowmaking on the changes in snow reliability over the entire time period. This study fills several gaps in the scientific literature, but comes with limitations which deserve some discussion.

### 4.1. Limitations to the methods employed

The SAFRAN meteorological reanalysis (Vernay et al., 2019, submitted) used for this research is affected by changes in observation network density from 1961 to 2018 and changes to the numerical weather prediction models used as a guess for producing the reanalysis. This can potentially impair its ability to be used for trend analysis, as noted in Spandre et al. (2015) and Ménégoz et al. (2020). In particular, Vernay et al. (submitted) indicate that the magnitude of the wintertime temperature trend is probably underestimated in the SAFRAN dataset, owing to temporal heterogeneities in the input data to SAFRAN

and stronger temperature deviations to reference observations at the beginning of the time period. The magnitude of the trend on snow cover (strongly influenced by air temperature trends especially at lower elevation), is therefore potentially underestimated. However, we note that our analysis here is not based on direct trend analysis but rather on how frequently snow conditions, computed at the monthly scale, fall above or below a given threshold, which depends on the meteorological conditions throughout the entire winter seasons. We also note that several key features (snow scarce winter 1963-1963, low

snow conditions in the late 1980s and early 1990s, challenging snow conditions in 2001-2002, 2006-2007 and further recent years etc.) are aptly reproduced by the model chain hence providing confidence in its ability to characterize the main features of the meteorological conditions of the winters of the past. While further analysis of the time heterogeneity in the SAFRAN reanalysis remains needed and is currently investigated, we consider that, due to the fact that SAFRAN uses all available observations in a robust assimilation framework, it is a usable data source for this work.

Another limitation of this work, on the snow cover modelling part, is the fact that the Crocus snow cover model not only uses the same snow management configuration for all ski resorts, but also it uses the same one throughout the entire time





period. Yet, snowmaking technology and management strategies and tactics have evolved over the past decades (Morrison and Pickering, 2013, Wilson et al., 2018). However, taking into account other sources of uncertainty, and the fact that the time evolution of the snowmaking coverage in ski resorts is itself largely unknown, accounting for changes in snowmaking

practices and on the technological characteristics of snowmaking units appears out of reach and may be approached in future studies, requiring that further data is made available (or reconstructed from technical historical sources from ski resorts internal documentation, which does not always exist). We however consider that our study, which accounts for grooming and snowmaking and brings together a large amount of original information into a consistent analysis framework, is a relevant addition to the sparse literature on observed changes in ski resorts operating conditions, and provides relevant and

valuable results. Another limitation related to changes in time for some of the characteristics of the systems studied is the fact that we used a fixed number of ski resorts across time (even if some opened after 1961), with constant ski lifts equipment, corresponding to the end of the time period. This could cause some heterogeneities due to the fact that ski slopes are related, in our work, to the location of ski lifts, and if major changes in ski lifts had occurred since the early 2000s when snowmaking developed the most, inconsistencies would occur. However, we believe that our approach is relevant because

the modelling framework makes it possible to simplify this complex problem, to some extent, by setting some of its dimensions to a given value (here: the geometry of ski resorts), and because the main features of ski resorts have in fact little evolved since the early 2000s, so that changes in operating conditions are more influenced by the inception and increase in snowmaking, than by changes in ski resorts spatial organization, e.g. related to the opening of a new subdomain owing to the installation of a new ski lift.

The modelling approach employed to estimate the evolution of the snowmaking coverage rate has also some limitations. In our study, the increase of the snow reliability is only possible through a growth of the part of the ski area covered with snowmaking. However, snowmaking investments considered in our study may correspond to other strategies of ski lift operators. Our model assumes that the entire snowmaking investments always resulted in an increase of the ski slope area covered by snowmaking. Therefore we removed 13 ski resorts without any snowmaking coverage growth despite

investments on purpose. This indicates that snowmaking investment choices are more complex than those taken into account in our linear modelling. For instance, our approach does not represent investments made to optimize existing systems – e.g. the replacement of obsolete snowguns - or to increase the density of snow guns on slopes already equipped. Moreover, based on discussions with snowmakers, we know the ski industry has experienced an ongoing harmonisation of the calculation method of the snowmaking coverage rate for many years, including through the development of tools dedicated to the

snowmaking management. We can speculate that the more extended snowmaking systems are, the more they require investments for their maintenance. Considering our model, the constant relationship between investments and area equipped let us think that the increase of snowmaking coverage may be overestimated. Our modelling also provides estimations considering the technology as remaining constant. As indicated above on the snow cover modelling side, we did not implement the potential effects of technological progress, particularly for recent snowmaking systems, that could lead to

yield increases. Similarly, we did not consider variations of investment costs in snowmaking over time, especially the



potential decrease of investment costs to cover ski slopes as snowmaking systems become increasingly popular or the influence of the snowmaking industry consolidation and its consequences on the prices.

Altogether, and despite its limitations, we believe that our study provides original and relevant information on observed changes in ski resorts operating conditions over climate time scales, contributing to filling a critical gap in the literature. We

also note that many of these limitations can only be waived by an increasing availability of data relevant to snow management in ski resorts, making it easier, if not even possible, to quantify the impact of climate change on their operating conditions, and, as a result, better understand their sensitivity to changing climate conditions and contributing to climate change adaptation in this domain (Spandre et al., 2019, Gerbaux et al., 2020, Berard-Chenu et al., 2020, Morin et al., 2021).

### 4.2.    Quantification of the impact of climate change on snow reliability, with and without snowmaking

This study provides an original appraisal of the time evolution of snow cover reliability, based on a detailed modelling framework enabling the computation of a resort-level snow reliability indicator, spanning the time period from 1961 to 2018 for multiple ski resorts. Our results indicate that, over the past 60 years in 16 ski resorts in Savoie, the climate conditions have become increasingly challenging for natural snow cover of ski resorts, especially for ski resorts with a significant fraction of their surface area below approx. 2000 m. This is particularly the case in early (November, December) and late

(March, April) season, with contrasting patterns depending on the fraction of the ski resort above 2000 m elevation. The results also indicate that not only natural snow cover reliability and its changes, but also the benefits of snowmaking in terms of snow reliability, significantly vary between ski resorts. While at the core of the winter season the snowmaking deployment trajectory has counterbalanced the decrease in natural snow cover reliability for ski resorts with a low elevation domain, and in most cases increased snow cover reliability, in early season and late season snowmaking development has not

counterbalanced the snow cover declining trend. In this respect, our results fully corroborate, in the case of the Northern French Alps, the findings highlighted in the IPCC SROCC Summary for Policymakers (IPCC 2019, in press) that while "[i]n nearly all high mountain areas, the depth, extent and duration of snow cover have declined over recent decades, especially at lower elevation (high confidence)" and that "[t]ourism and recreation, including ski [...] tourism [...] have [...] been negatively impacted in many mountain regions (medium confidence)", and "[i]n some places, artificial snowmaking has

reduced negative impacts on ski tourism (medium confidence)". Past impacts of climate change on natural snow cover reliability in ski resorts depends more on the indicator chosen to quantify the impacts and on resorts characteristics than on their mountain area they belong to. Indeed the added-value of snow management and snowmaking to reduce these impacts is also heterogeneous across ski resorts within a given mountain area. In this sense, while aggregation of results at the scale of an entire mountain range (e.g. Spandre et al., 2019 for the French Alps) makes it possible to provide a compact and general

picture of the impact of climate change, past or future, on the supply side of the ski tourism industry and raises awareness at the sectoral scale on climate change impacts and risks, it provides a lumped message, which is not applicable at the scale of individual ski resorts, although this is most often the most relevant governance level for mountain tourism development. Previous studies addressing large geographical domains have generally recognized that studies at the individual scale would





be most appropriate to inform deliberation and decision at the local scale, prompting for more detailed study at the scale of
individual ski resorts (e.g. Gerbaux et al., 2020; Steiger et al., 2019). The present study reinforces this position, and
reinforces the need to exercise extreme caution when attempting to derive general statements, at the local or regional levels,
about the evolution of ski resorts operating conditions. Furthermore, our results indicate that the trends in natural or managed
snow cover reliability strongly depend on the period of the year (at the monthly scale), which supports the need to carefully
choose relevant indicators depending on the ski resort considered (Abegg et al., 2020) : depending on their business model,
size, elevation range and climate settings, key periods for ski tourism can be different depending on the ski resort under
consideration, and the search for a single, universally applicable snow reliability indicator applicable to ski resorts of all
sizes and settings, is probably elusive.

### 4.3.   Can the climate change risk reduction factor due to snowmaking implementation be calculated?

Besides our analysis of the trends in natural and managed snow cover reliability in ski resorts in the Northern French Alps,
which we show varies across ski resorts due to differences in geographical settings (in particular the elevation distribution)
and pace of snowmaking equipment investments over the past decades, our analysis provides some of the information
required to assess whether snowmaking has indeed decreased the overall climate risk faced by ski resorts, due to climate
variability leading to the occasional occurrence of particularly unfavourable meteorological and snow conditions at the scale
of one year. In a context where both the climate and the extent to which snowmaking is implemented have changed, it is
virtually impossible to assess the partitioning of the climate change risk to its different factors. Still, *Domaines Skiables de
France*, the national organization bringing together French ski resorts, and the National Association of Mayors of Mountain
tourism villages (ANMSM) have stated that, thanks to snow management (grooming and snowmaking) and slope
preparation, the risk induced by a given level of climate hazard has been divided by a factor three over the past 25 years
(Domaines Skiables de France and ANMSM, 2015). The modelling approach taken in the present study allowed us to assess
how the current level of snowmaking coverage would have behaved against the situation which prevailed, for example,
during the notoriously challenging winter seasons in the late 1980s and early 1990s, er even the winter 1963-1964 which is,
for our study domain, the most challenging winter season, in terms of snow conditions, of the entire record from 1961 to
2018.  Figure 3, 5 and 6 show, for example, absolute values and differences to a no-snowmaking situation, in terms of annual
and monthly snow cover reliability, including these challenging winters. Figure 4 and 7 show the relationship between
reliability indicators accounting for a fixed, 2018-level snowmaking coverage, and the snow cover reliability values obtained
without snowmaking. This figures show that challenging winters without snowmaking remain challenging when accounting
for snowmaking at current snowmaking equipment levels, although in many cases the snow conditions are improved, but
never, especially in early (November and December) and late (March, April) season, to the extent when snowmaking fully
compensates for challenging natural snow conditions. In addition, the evidence provided in this study shows that the snow
reliability benefits of snowmaking for challenging winter seasons, i.e. its capacity to compensate for challenging
meteorological and natural snow conditions, is strongly heterogeneous across ski resorts due to their differences in





geographical settings (in particular their elevation range), but also the key periods during the winter season which are most impactful on their economic results. Providing a single factor for the climate risk reduction due to snowmaking across the entire ski industry in a given region would require that a single indicator can be applied to all ski resorts uniformly, and
furthermore that this indicator would be directly related with their economic performance in order to apply a risk approach meaningfully and compute the risk reduction factor related to a given snowmaking equipment level. While our modelling framework makes it possible to enable further quantifications in the future along these lines, our results suggest that the climate change risk reduction factor for the past period from 1961 to 2018 will undoubtedly be different for each ski resort, so that seeking a single climate change risk reduction factor for the entire ski tourism industry is most probably elusive and
would mask out the strong heterogeneity in the climate change impact drivers (hazards), exposure and vulnerability of the individual elements (ski resorts) forming the backbone of the industry as a whole.

### 4.4. Past performance is not a guarantee of future results.

The past increases in snow reliability due to snowmaking identified for some ski resorts in our study are consistent with the faith in snowmaking technology emphasized by some ski industry stakeholder risk perceptions studies. Trawöger (2014)
mentioned a "technosalvation" belief among Austrian CEOs of cable car companies and in a similar way Bicknell & Mcmanus (2006) showed an "overwhelming cornucopian belief" regarding snowmaking technology among Australian CEOs of ski lift companies. Steiger et al. (2019) stated that this belief is a widely consensus in the ski industry whereby "with constantly improving snowmaking techniques and continued investment, [ski lift operators] are well prepared for climate challenges that lie ahead". The partial success of snowmaking in counteracting challenging natural snow conditions may
skew ski lift operators' risk perception and lead them to consider past performance to plan future returns of snowmaking in reducing the impacts due to climate change. Yet, future success of snowmaking as an adaptive measure is not a guarantee in increasingly challenging climate conditions and ski lift operators should not assume that snowmaking investments will continue to do well in the future simply because they performed satisfyingly over past decades. This is especially the case considering that the snow production demand will worsen in a warming climate (Hock et al., in press).

## 5. Conclusions and perspectives

Our contribution introduces several innovative elements in the assessment of past changes in natural and managed snow reliability in the ski industry since it is based on past changes and not indirectly derived from future climate change projections. We also explicitly take into account the changing snowmaking coverage at the resort level. In order to achieve this, we inferred the individual snowmaking dynamic of each ski resort based on their respective snowmaking investments.
We then quantified snow reliability changes at the monthly time scale and focused on challenging winter seasons. We showed that the frequency of challenging conditions for ski resort operation over the 1991-2018 period increased in November and February to April compared to the reference period 1961-1990. In general, snowmaking had a positive impact





on snow reliability, especially in December to January. If for the highest ski resorts, snowmaking improved snow reliability for the entire winter season, it did not counterbalance the decreasing trend in snow cover reliability for lower elevation ski resorts. Contrasting outcomes of snowmaking highlight the various degrees of vulnerability across ski resorts due to their geographical settings and their business model which affected their snowmaking deployment trajectory. Such a high heterogeneity in the ski industry advocates for climate change risk assessment carried out, whenever possible, at the scale of individual ski resorts. Despite limitations both on snow cover modelling and the modelling of the snowmaking dynamics, our study is an additional element in providing more precise assessments about past changes in ski resort operating conditions. However, further progress in taking into account operating condition pathways (for a larger number of ski resorts, or with refinements in accounting for changes in technology), including snowmaking equipment, will depend on the increased availability of reliable resort-level information in various ski tourism markets.

## 6. Funding information

Within the CDP-Trajectories framework, the PhD scholarship of L. Berard-Chenu is jointly funded by the French National Research Agency in the framework of the "Investissements d'avenir" program (ANR-15-IDEX-02) and Météo-France. CNRM/CEN and LESSEM belong to LabEX OSUG@2020. This project has received funding from the European Union's Horizon 2020 research and innovation program under grant agreement no. 730203

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
