# Peer review of "Past changes in natural and managed snow reliability of French Alps ski resorts from 1961 to 2019"

_The Cryosphere, 2021_

## Referee Comment (RC1)

**General comment**

This is an interesting and valuable manuscript. I fully agree with the authors that there is little research on past changes in natural and managed snow reliability in ski resorts (i.e. there is more research on future changes in snow reliability in the context of climate change). I am also fully aware of the data problem (lack of data!) and, therefore, I was happy to learn that more data is available in France (or at least in Savoie) than in many other ski tourism markets. Given the data, the methodology is appropriate. The manuscript clearly highlights an important feature, and that is the heterogeneity of ski areas or ski resorts – with regard to size, elevation ranges (to distinguish between ski slopes above and below 2000 m, for example, is very helpful), snowmaking development paths and, although noted in the margin only), business models. Recently, quite a few scholars suggested to pay more attention to this heterogeneity; the authors of the paper at hand did it – in a good way. Consequently, this manuscript is a valuable contribution/addition to the existing literature and worth to be published.

**Specific comments**

Line 30f.: "… explicit assessments of the impact of climate change on ski resorts operations, based on past observations, have remained limited (Beaudin & Huang, 2014, Hamilton et al., 2003). Here, you should add the analogue studies, e.g. Rutty, M., Scott, D., Johnson, P., Pons, M., Steiger, R., & Vilella, M. (2017). Using ski industry response to climatic variability to assess climate change risk: An analogue study in Eastern Canada. Tourism Management, 58, 196–204 or Steiger, R. (2011). The impact of snow scarcity on ski tourism: An analysis of the record warm season 2006/2007 in Tyrol (Austria). Tourism Review, 66(3), 4–13.

Line 38ff.: "The strongest evidence on climate change impacts to ski tourism has therefore been inferred primarily from future climate change projections, which is then used to interpret past and present situations, in the absence of solid studies assessing impacts based on past observations." This is a strong statement, and open to debate. I refer to the sensibility analyses and the risk perception studies (see Steiger et al. 2019 for an overview) and recommend to reconsider the wording/phrasing.

Table 1: There is another figure often seen in the literature, although referring to investment costs per km (and not hectare): investment costs of approx. 1 million Swiss Francs per km ski slope (see www.seilbahnen.org and look for l'enneigement technique).

Fig. 4 (and therefore also Fig. 7):These are interesting figures. The corresponding explanation in the text, however, is very brief. For a better and/or faster understanding of these figures, it is suggested to add some explanatory text.

Further, in line 267 (referring to Fig. 5 and 6), you write about "… the lower values at the beginning and end of the winter season …". The lower values at the beginning of the season are clear. The lower values at the end of the season are not, maybe in April but this is not shown in the figures (and in March, the values are high).

Discussion (and limitations) is well done. In chapter 4.4 you could add Abegg, B., Steiger, R. & Trawöger, L. (2017). Resilience and perceptions of problems in Alpine regions. In R. W. Butler (Ed.), Tourism and resilience (pp. 105–117). Wallingford: CABI Publications. Conclusions and perspectives, though, are very brief. I would have expected to read more about the wider consequences of your research (e.g. the feasibility of future snowmaking investments in low and middle altitudes). And, apart from more company-/site-specific data is needed to refine this kind of research, what are exactly the perspectives, or more precisely, what is the research outlook?

**Some technical corrections**

General point: I am not a native English speaker but I think there is room for improvement in the use of the language.

- Line 34: led (instead of lead)
- Line 73: … snow cover simulation  using …
- Line 83: what do exactly mean by ski lift maintenance? Replacement of old ski lift by new ones?
- Line 110: "… it is not fully certain …" – this is a bit cryptic, please clarify.
- Fig. 3: divide into top, middle and bottom graph (and not into top, bottom and last row)
- Line 233: in 2018 (not: en)
- Line 247: it is with grooming only (and not without grooming), right?
- Line 456: These figures (not: this)

---

## Author Comment (AC1)

We thank Bruno Abegg for his helpful and constructive comments. The original comments of the reviewer are in blue. Our replies are in black.

**Bruno Abegg (Referee) #1**

**General comment**

This is an interesting and valuable manuscript. I fully agree with the authors that there is little research on past changes in natural and managed snow reliability in ski resorts (i.e. there is more research on future changes in snow reliability in the context of climate change). I am also fully aware of the data problem (lack of data!) and, therefore, I was happy to learn that more data is available in France (or at least in Savoie) than in many other ski tourism markets. Given the data, the methodology is appropriate. The manuscript clearly highlights an important feature, and that is the heterogeneity of ski areas or ski resorts – with regard to size, elevation ranges (to distinguish between ski slopes above and below 2000 m, for example, is very helpful), snowmaking development paths and, although noted in the margin only), business models. Recently, quite a few scholars suggested to pay more attention to this heterogeneity; the authors of the paper at hand did it – in a good way. Consequently, this manuscript is a valuable contribution/addition to the existing literature and worth to be published.

>>>> We thank B. Abegg for the positive general comments and helpful specific comments. We went through the comments and complemented and refined the manuscript accordingly, as described below.

**Specific comments**

Line 30f.: "… explicit assessments of the impact of climate change on ski resorts operations, based on past observations, have remained limited (Beaudin & Huang, 2014, Hamilton et al.,2003). Here, you should add the analogue studies, e.g. Rutty, M., Scott, D., Johnson, P., Pons, M., Steiger, R., & Vilella, M. (2017). *Using ski industry response to climatic variability to assess climate change risk: An analogue study in Eastern Canada*. Tourism Management, 58, 196–204 or Steiger, R. (2011). *The impact of snow scarcity on ski tourism: An analysis of the record warm season 2006/2007 in Tyrol (Austria).* Tourism Review, 66(3), 4–13.

>>>> We added both references mentioned, explicitly mentioning however that they correspond to analogue studies focusing on specific years rather than addressing past trends explicitly.

Line 38ff.: "The strongest evidence on climate change impacts to ski tourism has therefore been inferred primarily from future climate change projections, which is then used to interpret past and present situations, in the absence of solid studies assessing impacts based on past observations." This is a strong statement, and open to debate.

I refer to the sensibility analyses and the risk perception studies (see Steiger et al. 2019 for an overview) and recommend to reconsider the wording/phrasing.

>>>> We have rephrased this statement to avoid over-interpreting it, see below:

"*In fact, a substantial body of evidence on climate change impacts to ski tourism has been inferred primarily from future climate change projections rather than on the analysis of past changes in snow conditions*"

Table 1: There is another figure often seen in the literature, although referring to investment costs per km (and not hectare): investment costs of approx. 1 million Swiss Francs per km ski slope (see www.seilbahnen.org and look for "l'enneigement technique").

>>> We have added a reference to this information in the text:

"*Furger (2002) and RTS* [Remontées mécaniques suisses]*, the professional association of the Swiss ski resorts operators (2011) mentioned a cost investment in Switzerland but expressed it per km of slope equipped and not per hectare as figures in Table 1. Furger (2002) estimated the investment costs of snowmaking of 1 million Swiss Francs per km of slope equipped based on a sample of ten ski resorts located in the canton of Vaud*".

Fig. 4 (and therefore also Fig. 7): These are interesting figures. The corresponding explanation in the text, however, is very brief. For a better and/or faster understanding of these figures, it is suggested to add some explanatory text.

>>> We have expanded the description of the results of Figure 4. The description of Figure 7 was already more detailed and was therefore not expanded.

Further, in line 267 (referring to Fig. 5 and 6), you write about "… the lower values at the beginning and end of the winter season …". The lower values at the beginning of the season are clear. The lower values at the end of the season are not, maybe in April but this is not shown in the figures (and in March, the values are high).

>>> We have added April to Fig. 5, 6 and 7

Discussion (and limitations) is well done. In chapter 4.4 you could add Abegg, B., Steiger, R. & Trawöger, L. (2017). Resilience and perceptions of problems in Alpine regions. In R. W.Butler (Ed.), Tourism and resilience (pp. 105–117). Wallingford: CABI Publications.

>>>> We added the reference recommended since it mentions the optimistic confidence towards snowmaking to face climate change impacts in the ski tourism industry, see below:

"*According to Abegg et al. (2017) the faith placed in snowmaking by the ski tourism industry is one of the reasons why there remains a perception gap between scientific literature and some ski tourism industry stakeholders regarding climate change upcoming challenges.*"

Conclusions and perspectives, though, are very brief. I would have expected to read more about the wider consequences of your research (e.g. the feasibility of future snowmaking investments in low and middle altitudes). And, apart from more company-/site-specific data is needed to refine this kind of research, what are exactly the perspectives, or more precisely, what is the research outlook?

>>>> We have added further information about the wider implications of this work and perspectives, see below:

"*Nevertheless, our results pave the way for further studies addressing not only the detection and attribution of past changes and impacts of climate change on social-ecological systems, which requires analyzing not only changes in climatic-impact drivers but actual impacts, taking into account all the components of climate change risk (climatic-impact drivers, exposure and vulnerability).*
*Beyond these methodological perspectives, our work opens the way to broader investigations into the consequences of the development of snowmaking facilities in mountain regions. Steiger et al (2019) mentioned the challenge of a better understanding of ski tourism path dependency. We speculate that initial gains provided by snowmaking can foster the pursuit of these investments and embark ski tourism stakeholders on a path dependency. Beyond a general trend of snowmaking investments in Savoie, we pointed out that individual situations greatly differ in terms of snowmaking equipment dynamics and snow reliability outcomes, depending on ski resorts particularities. While this study explicitly considers geographical characteristics, it still under-estimates the potential influence of business models, which plays a key role for climate change vulnerability, with implications for the climate change adaptation strategy at the individual ski resort level, and warrants further investigations.*"

**Some technical corrections**
General point: I am not a native English speaker but I think there is room for improvement in the use of the language.
- Line 34: led (instead of lead)
>>>> Corrected

- Line 73: … snow cover simulation produced using …

>>>> Corrected

- Line 83: what do exactly mean by ski lift maintenance? Replacement of old ski lift by new ones?

>>>> Ski lift maintenance is not the replacement of old ski lifts by new ones. The ski lift maintenance refers to the servicing or partial replacement of existing ski lifts. Since public authorities frequently conduct periodic inspections of ski lifts, ski lift operators have to do some maintenance investments: cable replacement, electric system upgrades, compliance investments, etc.

We clarified to: *Ski lift maintenance (i.e. servicing and replacement of parts on existing ski lifts and compliance investments)*

- Line 110: "… it is not fully certain …" – this is a bit cryptic, please clarify.

>>>> We rephrased to:

*"The comparison between the DSF snowmaking facilities rate in 2020 and the estimation of Spandre et al. is difficult since the latter only considers the French Alps and DSF does not provide any information regarding the representativeness of its sample and how the aggregated value was derived for all ski resorts in France".*

- Fig. 3: divide into top, middle and bottom graph (and not into top, bottom and last row)

>>>> Corrected

- Line 233: in 2018 (not: en)

>>>> Corrected

- Line 247: it is with grooming only (and not without grooming), right?

>>>> That is correct, we rephrased the sentence to: "the snow cover reliability with grooming (no snowmaking, x-axis),"

- Line 456: These figures (not: this)

>>>> Corrected

Beyond the changes described below in response to the comments, and changes in response to the feedback from Reviewer #2, note that we have performed some further updates to the manuscript in order to improve. In particular, we have extended the time period covered by the study from the winter 1960-1961 to the winter 2018-2019. Figures 8, 9 and 10 have been improved by sorting the ski resorts the other way around (higher elevation resorts are at the top, not at the bottom).

---

## Author Comment (AC2)

We thank Referee #2 for his/her time and effort and helpful and constructive comments. The original comments of the reviewers are in blue. Our replies are in black.

I assess the study to have a high potential to contribute to the ski climate literature by focusing on (1) historical data, (2) climate variability, and (3) compensation potential of snowmaking for adaptive capacities. I believe the manuscript is good for publication but it could discuss some of my suggestions along with some minor points to be corrected below:

>>>> We thank Referee 2 for the positive and helpful comments. We went through the comments in detail and complemented and refined the manuscript accordingly.

- The authors intuitively claim that "there is more literature regarding future projections than past observed impacts." Can this be quantified, e.g. based on the Steiger review? It should also be noted that most of those future projections studies do bear (yet most often implicitly) past observations/reanalyses at least at some of point of their modellings (e.g. validation, calibration, bias correction etc).

>>>> It is correct that several studies addressing future climate change impacts use past climate in the process (for adjustment/bias correction or evaluation of the modelling systems), however we maintain that studies addressing past trends in snow cover reliability are quite rare, as noted by Reviewer #1 Bruno Abegg. We also note that studies addressing the relationship between past snow cover conditions and the performance of the ski tourism industry in the past, while based on past observations, do not provide information about potential trends in snow cover reliability. In this sense, several of the studies referred to as "Climate sensitivity assessments" in the Steiger et al. (2019) review do not provide information about snow cover reliability trends. The classification of scientific studies in Steiger et al. (2019) cannot directly be used to perform such an analysis, although we note that a few studies only (most of them quoted in our discussion article) have explicitly analyzed trends in snow cover reliability, which is consistent with our statement. However, for better clarity, we have rephrased the first sentence of the abstract to : "*Snow reliability is a key climatic impact-driver for the ski tourism industry, although there are only few studies addressing past changes in snow reliability in ski resorts accounting for snow management practices (grooming and snowmaking, in particular).*"

- I am not sure if the focus should be on the "developed" countries in the introduction especially now that we have China at the forefront of installing snowmaking systems and even promising an entire Olympic based on this technic.

>>>> We agree here and changed the sentence to:
"*Ski tourism is a major socio-economic component of mountainous regions for many countries around the world (Vanat, 2020).*"

- "Based on interviews with ski resort managers and ski tourists, several studies": Which "several" studies? You add a couple more in section 4.4, but is there any more to make them "several"?

>>>> We added references to studies based on interviews of ski tourism stakeholders and tourists:

Morrison, C., & Pickering, C. M. (2013) *Perceptions of climate change impacts, adaptation and limits to adaptation in the Australian Alps : The ski-tourism industry and key stakeholders.* Journal of Sustainable Tourism, 21(2), 173-191
Bicknell, S., & McManus, P. (2006) *The Canary in the Coalmine : Australian Ski Resorts and their Response to Climate Change.* Geographical Research, 44(4), 386-400.

- How do you justify the 25 cm (compared to the more commonly used 30 cm) threshold in your modelling? Is it field informed maybe for this particular case of the Savoie?

>>>> In fact, our analysis is based on a threshold in terms of snow mass (snow water equivalent) of 100 kg m$^{-2}$, which indeed corresponds to 25 cm of snow with a density of 400 kg m$^{-3}$ - but this number is only provided as an indication of what 100 kg m$^{-2}$ corresponds to. Our focus on snow mass, rather than snow depth, alleviates the influence on snow grooming on the snow depth (through compaction), and provides a fairer comparison between simulations with and without snow grooming. However, as we focus here on relative changes in snow cover reliability indices across the time periods considered, the sensitivity of our results to the exact threshold value used here is low, and very similar results would be obtained using a slightly different threshold (e.g. 100 or 120 kg m$^{-2}$ snow water equivalent).

- Please run a grammer check for Fig. 1.

>>>> We have thoroughly checked the wording in Fig. 1

- Lastly I would like to suggest two papers for your consideration as they may support your departure and enrich the discussions: Firstly, in a Swiss case, Gonseth (2013, Climatic Change) concludes that "an increase in the snowmaking percentage coverage from 30 % of the total length of ski runs to 50 % could counteract a 42 % increase in the natural snow conditions' variability" - a point to add to your discussions of snowmaking's added value to adaptive capacities. Secondly, in terms of a historical approach to climate variability, you can compare your innovative Q20 method and results to Mayer

et al. (2018, Sustainabiltiy) study which uses the ARCH/GARCH model to determine a significantly volatile historical visitation pattern, attributable to weather/climate variability, in the case of an Austrian glacier ski area.

>>>> We added the reference recommended of Gonseth's study (2013) in the discussion part since it mentions the snow reliability gains with snowmaking but also the diminishing returns to snowmaking investments. See below,

"*Our results are consistent with Gonseth (2013) who highlighted snow reliability gains with snowmaking. However this study stressed diminishing returns to snowmaking investments, similarly to Falk and Vanat (2016). Gonseth (2013) also pointed out snow reliability gains expected with snowmaking remained based on a dual assumption of economic and technical feasibility under future meteorological conditions.*"

>>>> We also added a reference to Mayer et al. (2018) in the discussion part. This study illustrates the importance of weather indices (e.g. thermal comfort index) to analyze the impact of climate conditions on the ski tourism industry. See below,

"*Snow reliability index and its derivatives remain one indicator among others to analyze the effect of climate conditions on the operation of the ski tourism industry. For instance, Mayer et al. (2018) illustrated that thermal comfort index could have a more significant effect on ski demand than snow depth. Several microclimatic characteristics may influence both the operation and visitation of the ski resorts.*"

Beyond the changes described below in response to the comments, and changes in response to the feedback from Reviewer #1 Bruno Abegg, note that we have performed some further updates to the manuscript in order to improve. In particular, we have extended the time period covered by the study from the winter 1960-1961 to the winter 2018-2019. Figures 8, 9 and 10 have been improved by sorting the ski resorts the other way around (higher elevation resorts are at the top, not at the bottom).